# Forecasting Treatment Responses Over Time Using Recurrent Marginal Structural Networks

**Bryan Lim**
Department of Engineering Science
University of Oxford
`bryan.lim@eng.ox.ac.uk`

**Ahmed Alaa**
Electrical Engineering Department
University of California, Los Angeles
`ahmedmalaa@ucla.edu`

**Mihaela van der Schaar**
University of Oxford
and The Alan Turing Institute
`mschaar@turing.ac.uk`

## Abstract

Electronic health records provide a rich source of data for machine learning methods to learn dynamic treatment responses over time. However, any direct estimation is hampered by the presence of time-dependent confounding, where actions taken are dependent on time-varying variables related to the outcome of interest. Drawing inspiration from marginal structural models, a class of methods in epidemiology which use propensity weighting to adjust for time-dependent confounders, we introduce the Recurrent Marginal Structural Network - a sequence-to-sequence architecture for forecasting a patient's expected response to a series of planned treatments. Using simulations of a state-of-the-art pharmacokinetic-pharmacodynamic (PK-PD) model of tumor growth [12], we demonstrate the ability of our network to accurately learn unbiased treatment responses from observational data – even under changes in the policy of treatment assignments – and performance gains over benchmarks.

## 1 Introduction

With the increasing prevalence of electronic health records, there has been much interest in the use of machine learning to estimate treatment effects directly from observational data [13, 41, 44, 2]. These records, collected over time as part of regular follow-ups, provide a more cost-effective method to gather insights on the effectiveness of past treatment regimens. While the majority of previous work focuses on the effects of interventions at a single point in time, observational data also captures information on complex time-dependent treatment scenarios, such as where the efficacy of treatments changes over time (e.g. drug resistance in cancer patients [40]), or where patients receive multiple interventions administered at different points in time (e.g. joint prescriptions of chemotherapy and radiotherapy [12]). As such, the ability to accurately estimate treatment effects over time would allow doctors to determine both the treatments to prescribe and the optimal time at which to administer them.

However, straightforward estimation in observational studies is hampered by the presence of time-dependent confounders, arising in cases where interventions are contingent on biomarkers whose value are affected by past treatments. For examples, asthma rescue drugs provide short-term rapid improvements to lung function measures, but are usually prescribed to patients with reduced lung function scores. As such, naïve methods can lead to the incorrect conclusion that the medication reduces lung function scores, contrary to the actual treatment effect [26]. Furthermore, [23] show

that the standard adjustments for causal inference, e.g. stratification, matching and propensity scoring [16], can introduce bias into the estimation in the presence of time-dependent confounding.

Marginal structural models (MSMs) are a class of methods commonly used in epidemiology to estimate time-dependent effects of exposure while adjusting for time-dependent confounders [15, 24, 19, 14]. Using the probability of a treatment assignment, conditioned on past exposures and covariate history, MSMs typically adopt inverse probability of treatment weighting (IPTW) to correct for bias in standard regression methods [22], re-constructing a 'pseudo-population' from the observational dataset to similar to that of a randomized clinical trial. However, the effectiveness of bias correction is dependent on a correct specification of the conditional probability of treatment assignment, which is difficult to do in practice given the complexity of treatment planning. In standard MSMs, IPTWs are produced using pooled logistic regression, which makes strong assumptions on the form of the conditional probability distribution. This also requires one separate set of coefficients to be estimated per time-step and many models to be estimated for long trajectories.

In this paper, we propose a new deep learning model - which we refer to as Recurrent Marginal Structural Networks - to directly learn time-dependent treatment responses from observational data, based on in the marginal structural modeling framework. Our key contributions are as follows:

**Multi-step Prediction Using Sequence-to-sequence Architecture**    To forecast treatment responses at multiple time horizons in the future, we propose a new RNN architecture for multi-step prediction based on sequence-to-sequence architectures in natural language processing [36]. This comprises two halves, 1) an encoder RNN which learns representations for the patient's current clinical state, and 2) a decoder which is initialized using the encoder's final memory state and computes forward predictions given the intended treatment assignments. At run time, the R-MSN also allows for prediction horizons to be flexibly adjusted to match the intended treatment duration, by expanding or contracting the number of decoder units in the sequence-to-sequence model.

**Scenario Analysis for Complex Treatment Regimens**    Treatment planning in clinical settings is often based on the interaction of numerous variables - including 1) the desired outcomes for a patient (e.g. survival improvement or comorbidity risk reduction), 2) the treatments to assign (e.g. binary interventions or continuous dosages), and 3) the length of treatment affected by both number and duration of interventions. The R-MSN naturally encapsulates this by using multi-input/output RNNs, which can be configured to have multiples treatments and targets of different forms (e.g. continuous or discrete). Different sequences of treatments can also be evaluated using the sequence-to-sequence architecture of the network. Moreover, given the susceptibility of IPTWs to model misspecification, the R-MSN uses Long-short Term Memory units (LSTMs) to compute the probabilities required for propensity weighting. Combining these aspects together, the R-MSN is able to help clinicians evaluate the projected outcome of a complex treatment scenario – providing timely clinical decision support and helping them customize a treatment regimen to the patient. A example of scenario analysis for different cancer treatment regimens is shown in Figure 1, with the expected response of tumor growth to no treatment, chemotherapy and radiotherapy shown.

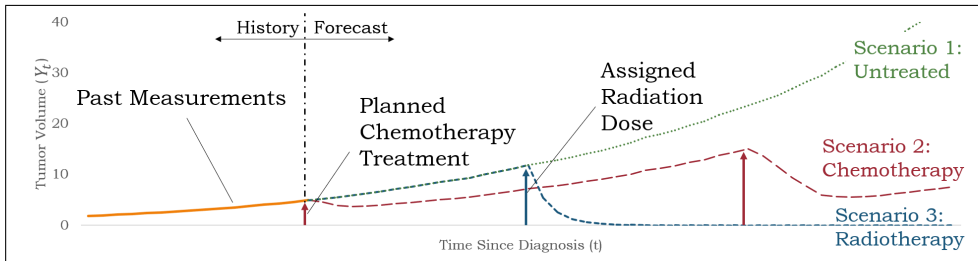

Figure 1: Forecasting Tumor Growth Under Multiple Treatment Scenarios

## 2   Related Works

Given the diversity of literature on causal inference, we focus on works associated with time-dependent treatment responses and deep learning here, with a wider survey in Appendix A.

**G-computation and Structural Models.** Counterfactual inference under time-dependent confounding has been extensively studied in the epidemiology literature, particularly in the seminal works of Robins [30, 31, 16]. Methods in this area can be categorized into 3 groups: models based on the G-computation formula, structural nested mean models, and marginal structural models [8]. While all these models provide strong theoretical foundations on the adjustments for time-dependent confounding, their prediction models are typically based on linear or logistic regression. These models would be misspecified when either the outcomes or the treatment policy exhibit complex dependencies on the covariate history.

**Potential Outcomes with Longitudinal Data.** Bayesian nonparametric models have been proposed to estimate the effects of both single [32, 33, 43, 34] and joint treatment assignments [35] over time. These methods use Gaussian processes (GPs) to model the baseline progression, which can estimate the treatment effects at multiple points in the future. However, some limitations do exist. Firstly, to aid in calibration, most Bayesian methods make strong assumptions on model structure - such as 1) independent baseline progression and treatment response components [43, 35], and 2) the lack of heterogeneous effects, by either omitting baseline covariates (e.g. genetic or demographic information) [34, 33] or incorporating them as linear components [43, 35]. Recurrent neural networks (RNNs) avoid the need for any explicit model specifications, with the networks learning these relationships directly from the data. Secondly, inference with Bayesian models can be computationally complex, making them difficult to scale. This arises from the use of Markov Chain-Monte Carlo sampling for g-computation, and the use of sparse GPs that have at least $O(NM^2)$ complexity, where $N$ and $M$ are the number of observations and inducing points respectively [39]. From this perspective, RNNs have the benefit of scalability and update their internal states with new observations as they arrive. Lastly, apart from [35] which we evaluate in Section 5, existing models do not consider treatment responses for combined interventions and multiple targets. This is handled naturally in our network by using multi-input/multi-output RNN architectures.

**Deep Learning for Causal Inference.** Deep learning has also been used to estimate individualized treatment effects for a single intervention at a fixed time, using instrumental variable approaches [13], generative adversarial networks [44] and multi-task architectures [3]. To the best of our knowledge, ours is the first deep learning method for time-dependent effects and establishes a framework to use existing RNN architectures for treatment response estimation.

## 3 Problem Definition

Let $\mathbf{Y}_{t,i} = [Y_{t,i}(1), \ldots, Y_{t,i}(\Omega_y)]$ be a vector of $\Omega_y$ observed outcomes for patient i at time $t$, $\mathbf{A}_{t,i} = [A_{t,i}(1), \ldots, A_{t,i}(\Omega_a)]$ a vector of actual treatment administered, $\mathbf{L}_{t,i} = [L_{t,i}(1), \ldots, L_{t,i}(\Omega_l)]$ time-dependent covariates and $\mathbf{X}_i = [X_i(1), \ldots, X_i(\Omega_v)]$ patient-specific static features. For notational simplicity, we will omit the subscript i going forward unless explicitly required.

**Treatment Responses Over Time** Determining an individual's response to a prescribed treatment can be characterized as learning a function $g(.)$ for the expected outcomes over a prediction horizon $\tau$, given an intended course of treatment and past observations, i.e.:

$$\mathbb{E}\left[\mathbf{Y}_{t+\tau}|a(t, \tau-1), \bar{\mathbf{H}}_t\right] = g(\tau, a(t, \tau-1), \bar{\mathbf{H}}_t) \tag{1}$$

where $g(.)$ represents a generic, possibly non-linear, function, $a(t, \tau-1) = (\mathbf{a}_t, \ldots \mathbf{a}_{t+\tau-1})$ is an *intended* sequence of treatments $\mathbf{a}_k$ from the current time until just before the outcome is observed, and $\bar{\mathbf{H}}_t = (\bar{\mathbf{L}}_t, \bar{\mathbf{A}}_{t-1}, \mathbf{X})$ is the patient's history with covariates $\bar{\mathbf{L}}_t = (\mathbf{L}_1, \ldots, \mathbf{L}_t)$ and actions $\bar{\mathbf{A}}_{t-1} = (\mathbf{A}_1, \ldots \mathbf{A}_{t-1})$.

**Inverse Probability of Treatment Weighting** Inverse probability of treatment weighting, extensively studied in marginal structural modeling to adjust for time-dependent confounding [22, 16, 15, 24, 26], with extensions to joint treatment assignments [19], censored observations [14] and continuous dosages [10]. We list the key results for our problem below, with a more thorough discussion in Appendix B.

The stabilized weights for joint treatment assignments [21] can be expressed as:

$$\mathbf{SW}(t, \tau) = \prod_{n=t}^{t+\tau} \frac{f(\mathbf{A}_n|\bar{\mathbf{A}}_{n-1})}{f(\mathbf{A}_n|\bar{\mathbf{H}}_n)} = \prod_{n=t}^{t+\tau} \frac{\prod_{k=1}^{\Omega_a} f(A_n(k)|\bar{\mathbf{A}}_{n-1})}{\prod_{k=1}^{\Omega_a} f(A_n(k)|\bar{\mathbf{H}}_n)} \tag{2}$$

where $f(.)$ is the probability mass function for discrete treatment applications, or the probability density function when continuous dosages are used [10]. We also note that $\bar{\mathbf{H}}_n$ contains both past treatments $\bar{\mathbf{A}}_{n-1}$ and potential confounders $\bar{\mathbf{L}}_n$. To account for censoring, we used the additional stabilized weights below:

$$SW^*(t,\tau) = \prod_{n=t}^{t+\tau} \frac{f(C_n = 0|\mathcal{T} > n, \bar{\mathbf{A}}_{\mathbf{n-1}})}{f(C_n = 0|\mathcal{T} > n, \bar{\mathbf{L}}_{n-1}, \bar{\mathbf{A}}_{\mathbf{n-1}}, \mathbf{X})} \tag{3}$$

where $C_n = 1$ denotes right censoring of the trajectory, and $\mathcal{T}$ is the time at which censoring occurs.

We also adopt the additional steps for stabilization proposed in [42], truncating stabilized weights at their 1st and 99th percentile values, and normalizing weights by their mean for a fixed prediction horizon, i.e. $\mathbf{S}\tilde{\mathbf{W}} = \mathbf{SW}_i(t,\tau)/\left(\sum_{i=1}^{I}\sum_{t=1}^{T_i}\mathbf{SW}_i(t,\tau)/N\right)$ where $I$ is the total number of patients, $T_i$ is the length of the patient's trajectory and N the total number of observations. Stabilized weights are then used to weight the loss contributions of each training observation, expressed in squared-errors terms below for continuous predictions:

$$e(i,t,\tau) = \mathbf{S}\tilde{\mathbf{W}}_i(t,\tau-1) \times S\tilde{W}_i^*(t,\tau-1) \times \|\mathbf{Y}_{t+\tau,i} - g(\tau, a(t,\tau-1), \bar{\mathbf{H}}_t)\|^2 \tag{4}$$

# 4 Recurrent Marginal Structural Networks

An MSM can be subdivided into two submodels, one modeling the IPTWs and the other estimating the treatment response itself. Adopting this framework, we use two sets of deep neural networks to build a Recurrent Marginal Structural Network (R-MSN) - 1) a set propensity networks to compute treatment probabilities used for IPTW, and 2) a prediction network used to determine the treatment response for a given set of planned interventions. Additional details on the algorithm can be found in Appendix E, with the source code uploaded onto GitHub[1].

## 4.1 Propensity Networks

From Equations 2 and 3, we can see that 4 key probability functions are required to calculate the stabilized weights. In all instances, probabilities are conditioned on the history of past observations ($\bar{\mathbf{A}}_{n-1}$ and $\bar{\mathbf{H}}_n$), making RNNs natural candidates to learn these functions.

Each probability function is parameterized with a different LSTM – collectively referred to as propensity networks – with action probabilities $f\left(\bar{\mathbf{A}}_n|\,.\,\right)$ generated jointly by a set of multi-target LSTMs and censoring probabilities $f\left(C_n = 0|\,.\,\right)$ by single output LSTMs. This also accounts for possible correlations between treatment assignments, for instance in treatment regimens where complementary drugs are prescribed together to combat different aspects of the same disease.

The flexibility of RNN architectures also allows for the modeling of treatment assignments with different forms. In simple cases with discrete treatment assignments, a standard LSTM with a sigmoid output layer can be used for binary treatment probabilities or a softmax layer for categorical ones. More complex architectures, such as variational RNNs [6], can be used to compute probabilities when treatments map to continuous dosages. To calculate the binary probabilities in the experiments in Section 5, LSTMs were fitted with tanh state activations and sigmoid outputs.

## 4.2 Prediction Network

The prediction network focuses on forecasting the treatment response of a patient, with time-dependent confounding accounted for using IPTWs from the propensity networks. Although standard RNNs can be used for one-step-ahead forecasts, actual treatments plans can be considerably more complex, with varying durations and number of interventions depending on the condition of the patient. To remove any restrictions on the prediction horizon or number of planned interventions, we propose the sequence-to-sequence architecture depicted in Figure 4.2. One key difference between our model and standard sequence-to-sequence (e.g.[36]) is that the last unit of the encoder is also used in making predictions for the first time step, in addition to the decoder units at further horizons. This allows the R-MSN to use all available information in making predictions, including the covariates

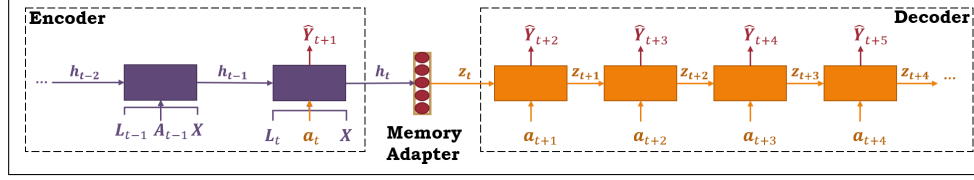

Figure 2: R-MSN Architecture for Multi-step Treatment Response Prediction

available at the current time step $t$. For the continuous predictions in Section 5, we used Exponential Linear Unit (ELU [7]) state activations and a linear output layer.

**Encoder** The goal of the encoder is to learn good representations for the patient's current clinical state, and we do so with a standard LSTM that makes one-step-ahead predictions of the outcome ($\hat{\mathbf{Y}}_{t+1}$) given observations of covariates and actual treatments. At the current follow-up time $t$, the encoder is also used in forecasting the expected response at $t+1$, as the latest covariate measurements $L_t$ are available to be fed into the LSTM along with the first planned treatment assignment.

**Decoder** While multi-step prediction can be performed by recursively feeding outputs into the inputs at the next time step, this would require output predictions for *all* covariates, with a high degree of accuracy to reduce error propagation through the network. Given that often only a small subset treatment outcomes are of interest, it would be desirable to forecast treatment responses on the basis of planned future actions alone. As such, the purpose of the decoder is to propagate the encoder representation forwards in time - using only the proposed treatment assignments and avoiding the need to forecast input covariates. This is achieved by training another LSTM that accepts only actions as inputs, but initializing the internal memory state of the first LSTM in the decoder sequence ($z_t$) using encoder representations. To allow for different state sizes in the encoder and decoder, encoder internal states ($h_t$) are passed through a single network layer with ELU activations, i.e. the memory adapter, before being initializing the decoder. As the network is made up of LSTM units, the internal states here refer to the concatenation of the cell and hidden states [17] of the LSTM.

## 4.3 Training Procedure

The training procedure for R-MSNs can be subdivided into the 3 training steps shown in Figure 3 - starting with the propensity networks, followed by the encoder, and ending with the decoder.

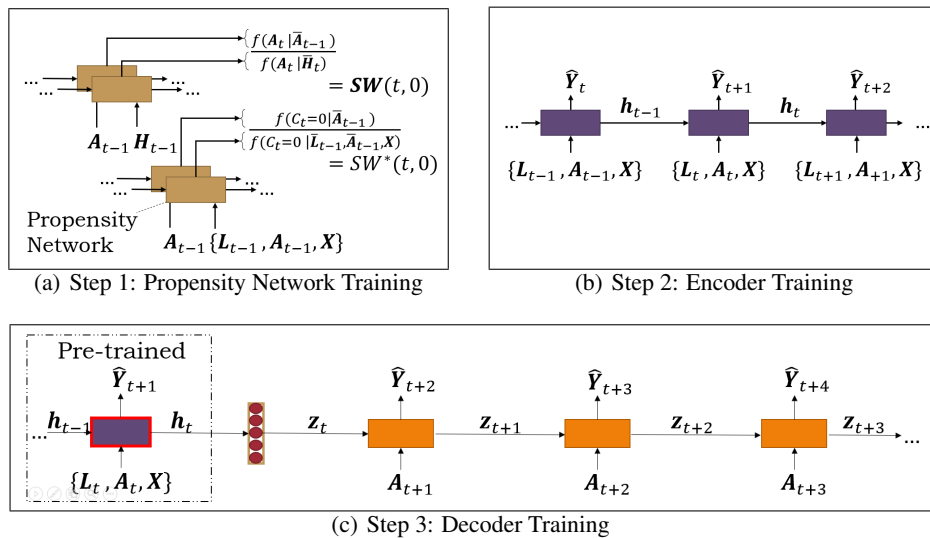

(a) Step 1: Propensity Network Training

(b) Step 2: Encoder Training

(c) Step 3: Decoder Training

Figure 3: Training Procedure for R-MSNs

**Step 1: Propensity Network Training** From Figure 3(a), each propensity network is first trained to estimate the probability of the treatment assigned at each time step, which is combined to compute $\mathbf{SW}(t,0)$ and $SW^*(t,0)$ at each time step. Stabilized weights for longer horizons can then be obtained from their cumulative product, i.e. $\mathbf{SW}(t,\tau) = \prod_{j=0}^{\tau} \mathbf{SW}(t+j,0)$. For tests in Section 5, propensity networks were trained using standard binary cross entropy loss, with treatment assignments and censoring treated as binary observations.

**Step 2: Encoder Training** Next, decoder and encoder training was divided into separate steps - accelerating learning by first training the encoder to learn representations of the patient's clinical state and then using the decoder to extrapolate them according to the intended treatment plan. As such, the encoder was trained to forecast standard one-step-ahead treatment response according to the structure in Figure 3(b), using all available information on treatments and covariates until the current time step. Upon completion, the encoder was used to perform a feed-forward pass over the training and validation data, extracting the internal states $\mathbf{h}_t$ for the final training step. As tests in Section 5 were performed for continuous outcomes, we express the loss function for the encoder as a weighted mean-squared error loss ($\mathcal{L}_{encoder}$ in Equation 5), although we note that this approach is compatible with other loss functions, e.g. cross entropy for discrete outcomes.

**Step 3: Decoder Training** Finally, the decoder and memory adapter were trained together based on the format in Figure 3(c). For a given patient, observations were batched into shorter sequences of up to $\tau_{max}$ steps, such that each sequence commencing at time $t$ is made up of $[\mathbf{h}_t, \{\mathbf{A}_{t+1}, \ldots, \mathbf{A}_{t+\tau_{max}-1}\}, \{\mathbf{Y}_{t+2}, \ldots, \mathbf{Y}_{t+\tau_{max}}\}]$. These were compiled for all patient-times and randomly grouped into minibatches to be used for backpropagation through time. For continuous predictions, the loss function for the decoder is ($\mathcal{L}_{decoder}$) can also be found in Equation 5.

$$\mathcal{L}_{encoder} = \sum_{i=1}^{I} \sum_{t=1}^{T_i} e(i,t,1) \qquad \mathcal{L}_{decoder} = \sum_{i=1}^{I} \sum_{t=1}^{T_i} \sum_{\tau=2}^{min(T_i-t,\tau_{max})} e(i,t,\tau) \qquad (5)$$

# 5 Experiments With Cancer Growth Simulation Model

## 5.1 Simulation Details

As confounding effects in real-world datasets are unknown a priori, methods for treatment response estimation are often evaluated using data simulations, where treatment application policies are explicitly modeled [34, 33, 35]. To ensure that our tests are fully reproducible and realistic from a medical perspective, we adopt the pharmacokinetic-pharmacodynamic (PK-PD) model of [12] - the state-of-the-art in treatment response modeling for non-small cell lung patients. The model features key characteristics present in actual lung cancer treatments, such as combined effects of chemo- and radiotherapy, cell repopulation after treatment, death/recovery of patients, and different staring distributions of tumor sizes based on the stage of cancer at diagnosis. On the whole, PK-PD models allow clinicians to explore hypotheses around dose-response relationships and propose optimal treatment schedules [5, 29, 11, 9, 1]. While we refer readers to [12] for the finer details of the model, such as specific priors used, we examine the overall structure of the model below to illustrate treatment-response relationships and how time-dependent confounding is introduced.

**PK-PD Model for Tumor Dynamics** We use a discrete-time model for tumor volume $V(t)$, where $t$ is the number of days since diagnosis:

$$V(t) = \left(1 + \underbrace{\rho \log(\frac{K}{V(t-1)})}_{\text{Tumor Growth}} - \underbrace{\beta_c C(t)}_{\text{Chemotherapy}} - \underbrace{(\alpha d(t) + \beta d(t)^2)}_{\text{Radiation}} + \underbrace{e_t}_{\text{Noise}} \right) V(t-1) \qquad (6)$$

where $\rho$, $K$, $\beta_c$, $\alpha$, $\beta$ are model parameters sampled for each patient according to prior distributions in [12]. A Gaussian noise term $e_t \sim N(0, 0.01^2)$ was added to account for randomness in the growth of the tumor. $d(t)$ is the dose of radiation applied at t, while drug concentration C(t) is modeled according to an exponential decay with a half life of 1 day, i.e.:

$$C(t) = \tilde{C}(t) + C(t-1)/2 \qquad (7)$$

where $\tilde{C}(t)$ is an new continuous dose of chemotherapy drugs applied at time t. To account for heterogeneous effects, we added static features to the simulation model by randomly subclassing

patients into 3 different groups, with each patient having a group label $S_i \in \{1, 2, 3\}$. This represents specific characteristics which affect with patient's response to chemotherapy and radiotherapy (e.g. by genetic factors [4]), which augment the prior means of $\beta_c$ and $\alpha$ according to:

$$\mu'_{\beta_c}(i) = \begin{cases} 1.1\mu_{\beta_c} \text{ , if } S_i = 3 \\ \mu_{\beta_c} \text{ , otherwise} \end{cases} \qquad \mu'_\alpha(i) = \begin{cases} 1.1\mu_\alpha \text{ , if } S_i = 1 \\ \mu_\alpha \text{ , otherwise} \end{cases} \qquad (8)$$

where $\mu_*$ are the mean parameters of [12], and $\mu'_*(i)$ those used to simulate patient i. We note that the value of $\beta$ is set in relation to $\alpha$, i.e. $\alpha/\beta = 10$, and would also be adjusted accordingly by $S_i$.

**Censoring Mechanisms** Patient censoring is incorporated by modeling 1) death when tumor diameters reach $D_{max} = 13 \ cm$ (or a volume of $V_{max} = 1150 \ cm^3$ assuming perfectly spherical tumors), 2) recovery determined by a Bernoulli process with recovery probability $p_t = \exp(-V_t)$, and 3) termination of observations after 60 days (administrative censoring).

**Treatment Assignment Policy** To introduce time-dependent confounders, we assume that chemotherapy prescriptions $A_c(t) \in \{0, 1\}$ and radiotherapy prescriptions $A_d(t) \in \{0, 1\}$ are Bernoulli random variables, with probabilities $p_c(t)$ and $p_d(t)$ respectively that are a functions of the tumor diameter:

$$p_c(t) = \sigma\left(\frac{\gamma_c}{D_{max}}(\bar{D}(t) - \theta_c)\right) \qquad p_d(t) = \sigma\left(\frac{\gamma_d}{D_{max}}(\bar{D}(t) - \theta_d)\right) \qquad (9)$$

where $\bar{D}(t)$ is the average tumor diameter over the last 15 days, $\sigma(.)$ is the sigmoid activation function, and $\theta_*$ and $\gamma_*$ are constant parameters. $\theta_*$ is fixed such that $\theta_c = \theta_d = D_{max}/2$, giving the model a 0.5 probability of treatment application exists when the tumor is half its maximum size. When treatments are applied, i.e. $A_c(t)$ or $A_d(t)$ is 1, chemotherapy is assumed to be administered in $5.0 \ mg/m^3$ doses of Vinblastine, and radiotherapy in $2.0 \ Gy$ fractions. $\gamma$ also controls the degree of time-dependent confounding - starting with no confounding at $\gamma = 0$, as treatment assignments are independent of the response variable, and an increase as $\gamma$ becomes larger.

## 5.2 Benchmarks

We evaluate the performance of R-MSNs against MSMs and Bayesian nonparametric models, focusing on its effectiveness in estimating unbiased treatment responses and its multi-step prediction performance. An overview of the models tested is summarized below:

**Standard Marginal Structural Models (MSM)** For the MSMs used in our investigations, we adopt similar approximations to [19, 14], encoding historical actions via cumulative sum of applied treatments, e.g. $cum(\bar{a}_c(t-1)) = \sum_{k=1}^{t-1} a_c(k)$, and covariate history using the previous observed value $V(t-1)$. The exact forms of the propensity and prediction models are in Appendix D.

**Bayesian Treatment Response Curves (BTRC)** We also benchmark our performance against the model of [35] - the state-of-the-art in forecasting multistep treatment responses for joint therapies with multiple outcomes. Given that the simulation model only has one target outcome, we also consider a simpler variant of the model without "shared" components, denoting this as the reduced BTRC (R-BTRC) model. This reduced parametrization was found to improve convergence during training, and additional details on calibration can be found in Appendix G.

**Recurrent Marginal Structural Networks (R-MSN)** R-MSNs were designed according to the description in Section 4, with full details on training and hyperparameter in Appendix F. To evaluate the effectiveness of the propensity networks, we also trained predictions networks using the IPTWs from the MSM, including this as an additional benchmark in Section 5.3 (Seq2Seq + Logistic).

## 5.3 Performance Evaluations

**Time-Dependent Confounding Adjustments** To investigate how well models learn unbiased treatment responses from observational data, we trained all models on simulations with $\gamma_c = \gamma_d = 10$ (biased policy) and examine the root-mean-squared errors (RMSEs) of one-step-ahead predictions as $\gamma_*$ is reduced. Both $\gamma_*$ parameters were set to be equal in this section for simplicity, i.e. $\gamma_c = \gamma_d = \gamma$. Using the simulation model in Section 5.1, we simulated 10,000 paths to be used for model training, 1,000 for validation data used in hyperparameter optimization, and another 1,000 for out-of-sample

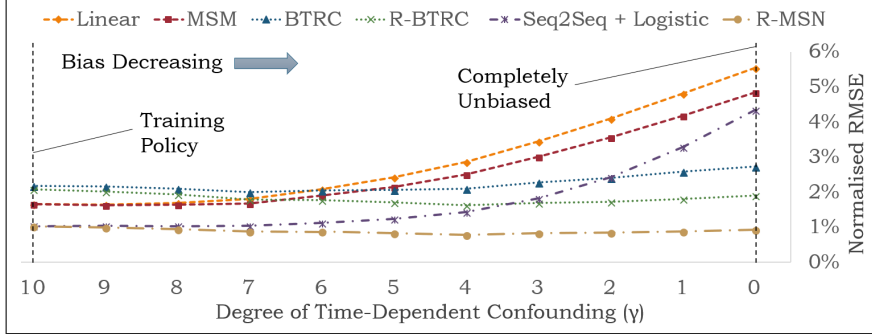

Figure 4: Normalized RMSEs for One-Step-Ahead Predictions

testing. For linear and MSM models, which do not have hyperparameters to optimized, we combined both training and validation datasets for model calibration.

Figure 4 shows the RMSE values of various models at different values of $\gamma$, with RMSEs normalized with $V_{max}$ and reported in percentage terms. Here, we focus on the main comparisons of interest – 1) linear models to provide a baseline on performance, 2) linear vs MSMs to evaluate traditional methods for IPTWs, 3) Seq2Seq + logistic IPTWs vs MSMs for the benefits of the Seq2Seq model, 4) R-MSN vs Seq2Seq + logistic to determine the improvements of our model and RNN-estimated IPTWs, and 5) BTRC/R-BTRC to benchmark against state-of-the-art methods. Additional results are also documented in Appendix C for reference.

From the graph, R-MSNs displayed the lowest RMSEs across all values of $\gamma$, decreasing slightly from a normalized RMSE of $1.02\%$ at $\gamma = 10$ to $0.92\%$ at $\gamma = 0$. Focusing on RMSEs at $\gamma = 0$, R-MSNs improve MSMs by $80.9\%$ and R-BTCs by $66.1\%$, demonstrating its effectiveness in learning unbiased treatment responses from confounded data. The propensity networks also improve unbiased treatment estimates by $78.7\%$ (R-MSN vs. Seq2Seq + Logistic), indicating the benefits of more flexible models for IPTW estimation. While the IPTWs of MSMs do provide small gains for linear models, linear models still exhibit the largest unbiased RMSE across all benchmarks - highlighting the limitations of linear models in estimating complex treatment responses. Bayesian models also perform consistently across $\gamma$, with normalized RMSEs for R-BTRC decreasing from $2.09\%$ to $1.91\%$ across $\gamma = 0$ to $10$, but were also observed to slightly underperform linear models on the training data itself. Part of this can potentially be attributed to model misspecification in the BTRC, which assumes that treatment responses are linear time-invariant and independent of the baseline progression. The differences in modeling assumptions can be seen from Equation 6, where chemotherapy and radiotherapy contributions are modeled as multiplicative with $V(t)$. This highlights the benefits of the data-driven nature of the R-MSN, which can flexibly learn treatment response models of different types.

**Multi-step Prediction Performance**    To evaluate the benefits of the sequence-to-sequence architecture, we report the normalized RMSEs for multi-step prediction in Table 1, using the best model of each category (R-MSN, MSM and R-BTRC). Once again, the R-MSN outperforms benchmarks for all timesteps, beating MSMs by $61\%$ on the training policy and $95\%$ for the unbiased one. While the R-BTRC does show improvements over MSMs for the unbiased treatment response, we also observe a slight underperformance versus MSMs on the training policy itself, highlighting the advantages of R-MSNs.

# 6   Conclusions

This paper introduces Recurrent Marginal Structural Networks - a novel learning approach for predicting unbiased treatment responses over time, grounded in the framework of marginal structural models. Networks are subdivided into two parts, a set of propensity networks to accurately compute the IPTWs, and a sequence-to-sequence architecture to predict responses using only a planned sequence of future actions. Using tests on a medically realistic simulation model, the R-MSN demonstrated performance improvements over traditional methods in epidemiology and the state-of-the-art models for joint treatment response prediction over multiple timesteps.

Table 1: Normalized RMSE for Various Prediction Horizons $\tau$

| | $\tau$ | 1 | 2 | 3 | 4 | 5 | Ave. % Decrease in RMSE vs MSMs |
|---|---|---|---|---|---|---|---|
| **Training** | MSM | 1.67% | 2.51% | 3.12% | 3.64% | 4.09% | - |
| **Policy** | R-BTRC | 2.09% | 2.85% | 3.50% | 4.07% | 4.58% | -32% (↑ RMSE) |
| $(\gamma_c = 10, \gamma_d = 10)$ | **R-MSN** | **1.02%** | **1.80%** | **1.90%** | **2.11%** | **2.46%** | **+61%** |
| **Unbiased** | MSM | 4.84% | 5.29% | 5.51% | 5.65% | 5.84% | - |
| **Assignment** | R-BTRC | 1.91% | 2.74% | 3.34% | 3.75% | 4.08% | +66% |
| $(\gamma_c = 0, \gamma_d = 0)$ | **R-MSN** | **0.92%** | **1.38%** | **1.30%** | **1.22%** | **1.14%** | **+95%** |
| **Unbiased** | MSM | 3.85% | 4.03% | 4.32% | 4.60% | 4.91% | - |
| **Radiotherapy** | R-BTRC | 1.74% | 1.68% | 2.14% | 2.54% | 2.91% | +74% |
| $(\gamma_c = 10, \gamma_d = 0)$ | **R-MSN** | **1.08%** | **1.66%** | **1.83%** | **1.98%** | **2.14%** | **+84%** |
| **Unbiased** | MSM | 1.84% | 2.65% | 3.09% | 3.44% | 3.83% | - |
| **Chemotherapy** | R-BTRC | 1.16% | 2.45% | 2.97% | 3.34% | 3.64% | +20% |
| $(\gamma_c = 0, \gamma_d = 10)$ | **R-MSN** | **0.65%** | **1.13%** | **1.05%** | **1.17%** | **1.31%** | **+87%** |

**Acknowledgments**

This research was supported by the Oxford-Man Institute of Quantitative Finance, the US Office of Naval Research (ONR), and the Alan Turing Institute.

## Footnotes

[1]`https://github.com/sjblim/rmsn_nips_2018`

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
