[Supplementary Material · nips_rmsn_supplementary.pdf]

# Appendix

## A  Extended Related Works

**Potential Outcomes with Cross-sectional Data.**  A simpler instantiation of the problem is to estimate the effect of a treatment applied to subjects in a (static) cross-sectional dataset. This problem has recently attracted a lot of attention in the machine learning community, and various interesting ideas were proposed to account for selection bias [3, 41, 44]. Unfortunately, most of these works cast the treatment effect estimation problem as one of learning under "covariate shift", where the goal is to learn a model for the outcomes that generalizes well to a population were treatments are randomly assigned to the subjects. Because of the sequential nature of the treatment assignment process in our setup, estimating treatment effects under time-dependent confounding cannot be similarly framed as a covariate shift problem, and hence the ideas developed in those works cannot be straightforwardly applied to our setup.

**Off-policy Evaluation.**  A closely related problem in the area of reinforcement learning is the problem of off-policy evaluation using retrospective observational data, also known as "logged bandit feedback" [38, 37, 18, 27, 28]. In this problem, the goal is to use sequences of states, actions and rewards generated by a decision-maker that operates under an unknown policy in order to estimate the expected reward of a given policy. In our setting, we focus on estimating a trajectory of outcomes given an application of a treatment (or a sequence of treatments) rather than estimating the average reward of a policy, and hence the "counterfactual risk minimization" framework in [37] would not result in optimal estimates in our setup. However, our learning model – with a different objective function– can be applied for the problem of off-policy evaluation.

## B  Background on Marginal Structural Models

In this section, we summarize the key relevant points from the seminal paper of Robins [31]. Without loss of generality, we consider the case of univariate treatments, response variables and baseline covariates here for simplicity.

Marginal structural models are typically considered in the context of follow-up studies, for example in patients with HIV [31]. Time in the study is typically measured in relation to a fixed starting point, such as the first follow-up date or time of diagnosis (i.e. $t = 1$). In such settings, marginal structural models are used to measure the average treatment effect conditioned on a series of potential actions and baseline covariate $V$ taken at the start of the study, expressed in the form:

$$\mathbb{E}\left[Y_\tau | a_1, \ldots, a_\tau, V\right] = r(a_1, \ldots, a_\tau, X; \Theta) \tag{10}$$

where $r(.)$ is a generic, typically linear, function with parameters $\Theta$.

**Time-Dependent Confounding.**  A full description of time-varying confounding can be found in [23], with formal definitions in [14]. Time-dependent confounding in observational studies arises as confounders have values which change over time - for example in cases where treatments are moderated based on the patient's response. A causal graph for 2-step study can be found in Figure 5, where $U$ denotes unmeasured factors. Note that $U_0, U_1$ do not have arrows to actions assignments, reflecting the assumption of no unmeasured confounding.

**Inverse Probability of Treatment Weighting**  From [31], under assumptions of no unmeasured confounding, positivity, and correct model specification, the stabilized IPTWs can be expressed as:

$$SW(\tau) = \prod_{n=0}^{\tau} \frac{f(A_n | \bar{A}_{n-1})}{f(A_n | \bar{A}_{n-1}, \bar{L}_n, X)} \tag{11}$$

Noting that $V$ is defined to be a subset of $L_0$ in [31]. Informally, they note the denominator to be conditional probability of a treatment assignment given past observations of treatment assignments and covariates and the numerator being that of treatment assignments alone, with the stabilized weights representing the incremental adjustment between the two.

Figure 5: Causal Graph of Time-dependent Confounding for 2-step Study

In real clinical settings, it is often desirable to determine the treatment response in relation to the current follow up time, given past information. As such, we consider trajectories in relation to the last follow-up time $t$, retaining the form of the stabilized weights of the MSM and using all past observations, i.e.

$$SW(t, \tau) = \prod_{n=t}^{t+\tau} \frac{f(A_n|\bar{A}_{n-1})}{f(A_n|\bar{A}_{n-1}, \bar{L}_n, X)} \tag{12}$$

## C  Additional Results for Experiments with Cancer Growth Simulation

Table 2 documents the full list of comparison for one-step-ahead predictions when tested for various $\gamma$, using different combinations of prediction and IPTW models.

| $\gamma =$ | 0 | 1 | 2 | 3 | 4 | 5 |
|---|---|---|---|---|---|---|
| Linear (No IPTWs) | 5.55% | 4.81% | 4.09% | 3.44% | 2.86% | 2.42% |
| MSM | 4.84% | 4.19% | 3.56% | 3.00% | 2.51% | 2.15% |
| MSM (LSTM IPTWs) | 4.26% | 3.68% | 3.13% | 2.64% | 2.22% | 1.95% |
| Seq2Seq (No IPTWs) | 1.52% | 1.39% | 1.28% | 1.23% | 1.17% | 1.17% |
| Seq2Seq (Logistic IPTWs) | 4.34% | 3.28% | 2.42% | 1.83% | 1.43% | 1.23% |
| R-MSN | 0.92% | 0.89% | 0.85% | 0.84% | 0.79% | 0.84% |
| BTRC | 2.73% | 2.59% | 2.42% | 2.28% | 2.11% | 2.07% |
| R-BTRC | 1.91% | 1.81% | 1.72% | 1.69% | 1.63% | 1.71% |
| $\gamma =$ | 6 | 7 | 8 | 9 | 10 | |
| Linear (No IPTWs) | 2.09% | 1.80% | 1.70% | 1.65% | 1.66% | |
| MSM | 1.90% | 1.68% | 1.64% | 1.64% | 1.67% | |
| MSM (LSTM IPTWs) | 1.77% | 1.61% | 1.62% | 1.64% | 1.69% | |
| Seq2Seq (No IPTWs) | 1.13% | 1.07% | 1.07% | 1.09% | 1.08% | |
| Seq2Seq (Logistic IPTW) | 1.12% | 1.04% | 1.04% | 1.05% | 1.04% | |
| R-MSN | 0.88% | 0.88% | 0.94% | 1.00% | 1.02% | |
| BTRC | 2.05% | 2.01% | 2.10% | 2.16% | 2.19% | |
| R-BTRC | 1.77% | 1.79% | 1.93% | 2.02% | 2.09% | |

Table 2: One-step-ahead Prediction Performance for Models Calibrated on $\gamma = 10$

## D   Marginal Structural Models for Cancer Simulation

The probabilities required for the IPTWs of the standard MSM in Section 5.2 can be described using logistic regression models with equations below:

$$f(A_t(k)|\bar{\mathbf{A}}_t) = \sigma\big(\omega_1^{(k)} \,(\sum_{n=0}^{t} \bar{A}_c(n-1)) + \omega_2^{(k)} \,(\sum_{n=0}^{t} \bar{A}_d(n-1)))\big) \tag{13}$$

$$f(A_t(k)|\bar{\mathbf{H}}_t) = \sigma\big(\omega_5^{(k)} \,(\sum_{n=0}^{t} \bar{A}_c(n-1)) + \omega_6^{(k)} \,\sum_{n=0}^{t}(\bar{A}_d(n-1))$$
$$+\omega_7^{(k)} \,V(t) + \omega_8^{(k)} \,V(t-1) + \omega_9^{(k)} \,S\big) \tag{14}$$

$$f(C_t = 0|\mathcal{T} > n, \bar{\mathbf{A}}_{n-1}) = \sigma\big(\omega_{10} \,(\sum_{n=0}^{t} \bar{A}_c(n-1)) + \omega_{11} \,\sum_{n=0}^{t}(\bar{A}_d(n-1)))\big) \tag{15}$$

$$f(C_t = 0|\mathcal{T} > n, \bar{\mathbf{L}}_{n-1}, \bar{\mathbf{A}}_{n-1}, \mathbf{X}) = \sigma\big(\omega_{12} \,(\sum_{n=0}^{t} \bar{A}_c(n-1)) + \omega_{13} \,(\sum_{n=0}^{t} \bar{A}_d(n-1))$$
$$+\omega_{14} \,V(t-1) + \omega_{15} \,S\big) \tag{16}$$

where $\sigma(.)$ the sigmoid function and $\omega_*$ are regression coefficients.

The regression model for prediction is given by:

$$g(\tau, a(t, \tau-1), \bar{\mathbf{H}}_t) = \beta_1 \,(\sum_{n=0}^{t} \bar{A}_c(n-1)) + \beta_2 \,(\sum_{n=0}^{t} \bar{A}_d(n-1))$$
$$+\beta_3 \,V(t) + \beta_4 \,V(t-1) + \beta_5 \,S \tag{17}$$

## E   Algorithm Description for R-MSNs

To provide additional clarity on the relationship between the propensity networks and the Seq2Seq model, the pseudocode in Algorithm 1 describes the training process mentioned in Section 4.3.

We first define function $r_{(.)} \,(.; \boldsymbol{\theta}_{(.)})$ to be RNN outputs given a vector of weights and hyperparameters $\boldsymbol{\theta}_{(.)}$. We refer the reader to Section 3 for more information on the functions in the MSM framework approximated by RNNs.

**Propensity Networks**   Components of the propensity networks are used to compute the IPTWs $\mathbf{SW}(t, \tau)$ and $SW^*(t, \tau)$ as defined in Equations 2 and 3 respectively. The probabilities in the numerators and denominators are taken to be outputs of the propensity networks as below:

$$f(\mathbf{A}_n|\bar{\mathbf{A}}_{n-1}) = r_{A1}(\mathbf{A}_n|\bar{\mathbf{A}}_{n-1}; \boldsymbol{\theta}_{A1}) \tag{18}$$

$$f(\mathbf{A}_n|\bar{\mathbf{H}}_n) = r_{A2}(\mathbf{A}_n|\bar{\mathbf{H}}_n; \boldsymbol{\theta}_{A2}) \tag{19}$$

$$f(C_n = 0|\mathcal{T} > n, \bar{\mathbf{A}}_{n-1}) = r_{C1}(\bar{\mathbf{A}}_{n-1}; \boldsymbol{\theta}_{C1}) \tag{20}$$

$$f(C_n = 0|\mathcal{T} > n, \bar{\mathbf{L}}_{n-1}, \bar{\mathbf{A}}_{n-1}, \mathbf{X}) = r_{C1}(\bar{\mathbf{L}}_{n-1}, \bar{\mathbf{A}}_{n-1}, \mathbf{X}; \boldsymbol{\theta}_{C2}) \tag{21}$$

**Encoder**   The encoder is also defined in a similar fashion below, with an additional function to output the internal states of the LSTM $\tilde{g}_{E1}(\bar{\mathbf{L}}_t, \bar{\mathbf{A}}_t, \mathbf{X}; \boldsymbol{\theta}_{E1})$. The encoder also computes the one-step-ahead predictions, i.e. $g(1, a(t, 0), \bar{\mathbf{H}}_t)$ as per Equation 1, which is to define the prediction error $e(i, t, 1)$ and encoder loss $\mathcal{L}_{encoder}$ – i.e.Equations 4 and 5 respectively.

$$g(1, a(t, 0), \bar{\mathbf{H}}_t) = r_E(\bar{\mathbf{L}}_t, \bar{\mathbf{A}}_t, \mathbf{X}; \boldsymbol{\theta}_E) \tag{22}$$

$$\mathbf{h}_t = \tilde{r}_E(\bar{\mathbf{L}}_t, \bar{\mathbf{A}}_t, \mathbf{X}; \boldsymbol{\theta}_E) \tag{23}$$

**Decoder** The decoder then uses the seq2seq architecture to project encoder states $\mathbf{h}_t$ forwards in time, incorporating planned future actions $\mathbf{a}_{t+\tau}$. This is also combined with the IPTWs to define the decoder loss $\mathcal{L}_{decoder}$ in Equation 5.

$$g(\tau, a(t, \tau - 1), \bar{\mathbf{H}}_t) = r_D(\mathbf{h}_t, a_{t+1}, \ldots, a_{t+\tau}; \boldsymbol{\theta}_D), \forall \tau > 1 \qquad (24)$$

---

**Algorithm 1** Training Process for R-MSN

---

**Input:** Training/Validation Data $\bar{\mathbf{L}}_{1:T}, \ \bar{\mathbf{A}}_{1:T}, \ X$
**Output:** Neural network weights and hyperparameters for:
    1) $\mathbf{SW}(t, \tau)$ networks: $\boldsymbol{\theta}_{A1}, \ \boldsymbol{\theta}_{A2}$
    2) $SW^*(t, \tau)$ networks: $\boldsymbol{\theta}_{C1}, \ \boldsymbol{\theta}_{C2}$
    3) Encoder network: $\boldsymbol{\theta}_{E1}, \ \boldsymbol{\theta}_{E2}$
    4) Decoder network: $\boldsymbol{\theta}_{D1}, \ \boldsymbol{\theta}_{D2}$
1:
2: **Step 1: Fit Propensity Networks**
3: $\boldsymbol{\theta}_{A1} \leftarrow \texttt{optimize}\Big( \sum_{n,i} \texttt{binary\_x\_entropy}(r_{A1}(\mathbf{A}_n(i)|\bar{\mathbf{A}}_{n-1}(i); \boldsymbol{\theta}_{A1}), \mathbf{A}_n(i)) \Big)$

4: $\boldsymbol{\theta}_{A2} \leftarrow \texttt{optimize}\Big( \sum_{n,i} \texttt{binary\_x\_entropy}(r_{A2}(\mathbf{A}_n(i)|\bar{\mathbf{H}}_n(i); \boldsymbol{\theta}_{A2}), \mathbf{A}_n(i)) \Big)$

5: $\boldsymbol{\theta}_{C1} \leftarrow \texttt{optimize}\Big( \sum_{n,i} \texttt{binary\_x\_entropy}(r_{C1}(\bar{\mathbf{A}}_{\mathbf{n-1}}(\mathbf{i}); \boldsymbol{\theta}_{C1}(i)), \mathbf{C}_n(i)) \Big)$

6: $\boldsymbol{\theta}_{C2} \leftarrow \texttt{optimize}\Big( \sum_{n,i} \texttt{binary\_x\_entropy}(r_{C2}(\bar{\mathbf{L}}_{n-1}(i), \bar{\mathbf{A}}_{n-1}(i), \mathbf{X}(i); \boldsymbol{\theta}_{C2}), \mathbf{C}_n(i)) \Big)$

7:
8: **Step 2: Generate IPTWs**
9: **for** patient $i = 1$ to $I$ **do**
10:    **for** $t = 1$ to $T$ **do**
11:       **for** $\tau = 1$ to $\tau_{max}$ **do**
12:          $\mathbf{SW}_i(t, \tau) \leftarrow \prod_{n=t}^{t+\tau} r_{A1}(\mathbf{A}_n(i)|\bar{\mathbf{A}}_{n-1}(i); \boldsymbol{\theta}_{A1}) \ / \ r_{A2}(\mathbf{A}_n(i)|\bar{\mathbf{H}}_n(i); \boldsymbol{\theta}_{A2})$
13:          $SW_i^*(t, \tau) \leftarrow \prod_{n=t}^{t+\tau} r_{C1}(\bar{\mathbf{A}}_{\mathbf{n-1}}(\mathbf{i}); \boldsymbol{\theta}_{C1}(i)) \ / \ r_{C2}(\bar{\mathbf{L}}_{n-1}(i), \bar{\mathbf{A}}_{n-1}(i), \mathbf{X}(i); \boldsymbol{\theta}_{C2})$
14:       **end for**
15:    **end for**
16: **end for**
17:
18: **Step 3: Fit Encoder**
19: $\boldsymbol{\theta}_E \leftarrow \texttt{optimize}\big(\mathcal{L}_{encoder}\big)$, as per Equation 5a
20:
21: **Step 4: Compute Encoder States** {Used to Initialize Decoder}
22: **for** patient $i = 1$ to $I$ **do**
23:    **for** $t = 1$ to $T$ **do**
24:       $\mathbf{h}_t(i) \leftarrow \tilde{g}_E(\bar{\mathbf{L}}_t(i), \bar{\mathbf{A}}_t(i), \mathbf{X}(i); \boldsymbol{\theta}_E)$
25:    **end for**
26: **end for**
27:
28: **Step 5: Fit Decoder**
29: $\boldsymbol{\theta}_D \leftarrow \texttt{optimize}\big(\mathcal{L}_{decoder}\big)$, as per Equation 5b

---

# F Hyperparameter Optimization for R-MSN

For the R-MSN, 10,000 simulated paths were used for backpropagation of the network (training data), and 1,000 simulated paths for hyperparameter optimization (validation data) - with another 1,000 for out-of-sample testing. Given the differences in state initialization requirements and data batching of the decoder, we report the hyperparameter optimization settings separately for the decoder. The optimal parameters of all networks can be found in Table 5.

**Settings for Propensity Networks and Encoder**    Hyperparameter optimization was performed using 50 iterations of random search, using the hyperparameter ranges in Table 3, and networks were trained using the ADAM optimizer [20]. For each set of sampled, simulation trajectories were grouped into B minibatches and networks were trained for a maximum of 100 epochs. LSTM state sizes were also defined in relation to the number of inputs for the network $C$.

Table 3: Hyperparameter Search Range for Propensity Networks and Encoder

|  | Hyperparameter Search Range |
| --- | --- |
| **Hyperparameter Search Iterations** | 50 |
| **Dropout Rate** | 0.1 , 0.2 , 0.3, 0.4, 0.5 |
| **State Size** | 0.5C, 1C, 2C, 3C, 4C |
| **Minibatch Size** | 64, 128, 256 |
| **Learning Rate** | 0.01, 0.005, 0.001 |
| **Max Gradient Norm** | 0.5, 1.0, 2.0 |

**Settings for Decoder**    To train the decoder, the data was reformatted into sequences of $(\mathbf{h}_t, \{\mathbf{L}_{t+1}, \ldots, \mathbf{L}_{t+\tau_{max}}\}, \{\mathbf{A}_t, \ldots, \mathbf{A}_{t+\tau_{max}}, \mathbf{X}\})$, such that each patient $i$ max $T_i$ contributions to the training dataset. Given the $T$-fold increase in the number of rows in the overall dataset, we made a few modifications to the range of hyperparameter search, including increasing the size of minibatches and reducing the learning rate and number of iterations of hyperparameter search. The full range of hyperparameter search can be found in Table 4 and networks are trained for maximum of 100 epochs as well.

Table 4: Hyperparameter Search Range for Decoder

|  | Hyperparameter Search Range |
| --- | --- |
| Iterations of Hyperparameter Search | 20 |
| Dropout Rate | 0.1 , 0.2 , 0.3, 0.4, 0.5 |
| State Size | 1C, 2C, 4C, 8C, 16C |
| Minibatch Size | 256, 512, 1024 |
| Learning Rate | 0.01, 0.005, 0.001, 0.0001 |
| Max Gradient Norm | 0.5, 1.0, 2.0, 4.0 |

Table 5: Optimal Hyperparameters for R-MSN

|  | Dropout Rate | State Size | Minibatch Size | Learning Rate | Max Norm |
| --- | --- | --- | --- | --- | --- |
| **Propensity Networks** |  |  |  |  |  |
| $f(\mathbf{A}_n\|\bar{\mathbf{A}}_{n-1})$ | 0.1 | 6 (3C) | 128 | 0.01 | 2.0 |
| $f(\mathbf{A}_n\|\bar{\tilde{H}}_n)$ | 0.1 | 16 (4C) | 64 | 0.01 | 1.0 |
| $f(C_n = 0\|\mathcal{T} > n, \bar{\mathbf{A}}_{\mathbf{n-1}})$ | 0.2 | 4 (2C) | 128 | 0.01 | 0.5 |
| $f(C_t = 0\|\mathcal{T} > n, \bar{\mathbf{L}}_{n-1}, \bar{\mathbf{A}}_{n-1}, \mathbf{X})$ | 0.1 | 16 (4C) | 64 | 0.01 | 2.0 |
| **Prediction Networks** |  |  |  |  |  |
| Encoder | 0.1 | 16 (4C) | 64 | 0.01 | 0.5 |
| Decoder + Memory Adapter | 0.1 | 16 (8C) | 512 | 0.001 | 4.0 |

# G   Hyperparameter Optimization for BTRC

The parameters of the BTRC were optimized using the maximum-a-posteriori (MAP) estimation, using the same prior for global parameters and approach defined in [35]. While the model was replicated as faithfully to the specifications as possible, two slight modifications were made to adapt it to our problem. Firstly, the sparse GP approximations were avoided to ensure that we had as much accuracy as possible - using Gaussian Process with full covariance matrices for the random effects components. Secondly, as our dataset was partitioned to ensure that patient observed in the training set were not present in the test set, this means that any patient-specific parameters learned would not be used in the testing set itself. As such, to avoid optimizing on the test set, we adopt the standard approach for prediction in generalized linear mixed models [25], using the average population parameters, i.e. the global MAP estimate, for prediction.

Hyperparameter optimization was performed using grid search on the optimizer settings defined in 6, and was performed for a maximum of 5000 epochs per configuration. As convergence was observed to be slow for a number of settings, we also trained a reduced form of the full BTRC model without the "shared" parameters (indicated by '-' in Table7) to reduce the number of parameters of the model. The optimal global hyperparameters and optimizer settings can be found in Table 7.

Table 6: Hyperparameter Grid for BTRC

|  | **Hyperparameter Search Range** |
| --- | --- |
| **Minibatch Size** | 2, 5, 10, 100, 500 |
| **Learning Rate** | $10^{-1}, 10^{-2}, 10^{-3}, 10^{-4}, 10^{-5}$ |

Table 7: MAP Estimates for BTRC

|  | **BTRC** | **R-BTRC** |
| --- | --- | --- |
| $\bar{\chi}_{chemo}, \bar{\chi}_{radio}$ | (-1.3729433, 0.065) | (-1.162, 0.007) |
| $\bar{\alpha}_{chemo}, \bar{\alpha}_{radio},$ $\bar{\alpha}_{chemo}^{(0)}, \bar{\alpha}_{radio}^{(0)}$ | (0.760, 0.367), (0.207, 0.490) | (0.547, 0.367), ( -, -) |
| $\bar{\beta}_{chemo}, \bar{\beta}_{radio},$ $\bar{\beta}_{chemo}^{(0)}, \bar{\beta}_{radio}^{(0)}$ | (0.595, 0.367), (0.204, 0.368) | (0.429, 0.368), ( -, -) |
| $\bar{\gamma}$ | -0.27 | -0.262 |
| $\bar{\omega}$ | -0.928 | - |
| $\bar{l}^g$ | 1.223 | - |
| $\bar{\kappa}$ | 0.786 | 0.867 |
| $\bar{l}^v$ | 1.092 | 1.151 |
| $\bar{\sigma}^2$ | 0.036 | 0.042 |
| Learning Rate | 0.001 | 0.001 |
| Minibatch Size | 100 | 100 |