[Reviews · NeurIPS 2018]

Reviewer 1



The authors rebuttal was very helpful and could clarify many questions. Especially, the novel contribution of learning a model for treatment effects over time made as improving our ratings. Treatment response forecasting always suffers from time-dependent confounders, making the exploration of causality very difficult. The standard approach to deal with this problem are Marginal Structural Models (MSM). However, these methods strongly rely on the accuracy of manual estimation steps. In this work, the authors present a new deep learning approach, embedding the concept of MSM into an LSTM framework. They could show that the deep learning approach outperformed the 5 alternative methods (including the state-of-the-art). The paper presents a very interesting approach to an important problem. It is based on profound theoretical knowledge about the specific problem and the network is designed with care. The extensive appendix is worth a comment, including theoretical background information, additional resources as well as details on the hyperparameter optimization. However, the work is application-driven and the learning methods itself are not new. Additionally, the information on the network architecture is insufficient for reimplementing the method and partially incomplete. I would highly suggest to include a sketch of the whole workflow, including all separate neural networks. All in all, the paper presents a profound work good results that are important for the community of treatment outcome forecasting, but the contribution to the field of neural learning itself is small. I highly recommend a submission as a survey in the relevant research area. In the following we discuss conceptual misunderstandings or unclear formulations which could be improved. On computing probabilities with neural networks: In page 2, LSTM, computing probabilities and the keyword robust are used in lines 64- 67, in one sentence, which gives the impression that LSTMs can do that which is actually not the case. Even for binary random variables only point estimates can be sampled which have to collected and post processed to compute uncertainty estimates. Also in page 4, the need to compute probabilities is discussed. For binary random variables, LSTM with a softmax output layer are used in the paper. The outlook on an extension using variational RNNs refers to a review paper that does not discuss rnns. Actually, computing prob. outputs in rnns is an active research area and I was surprised that the authors refer to existing solutions. Minors: Notation: Using upperscripts for the patient number Y_t^[i] would simplify the notation. Also using capital letters for matrizes and bold lower case letters for vectors should improve the readability. p. 4. line 130: are then used to p. 3 line 115-118: missing verb

Reviewer 2



Post-rebuttal comment: In their response, the authors answered my primary question about breaking apart their model (which consists of two RNN components) to identify more specifically where the gains of this model come from. They also explained how Figure 4 contains similar comparisons, and clarified my question about the gamma = 0 setting. I think the paper would benefit from a little more clarity around Figure 4 to minimise the confusion I experienced, but otherwise I remain confident in my assessment of the quality of the paper. I note also that their response to me also addressed one of the concerns of Reviewer 3. Summary of work: This work is focused on estimating treatment responses through time in the presence of time-dependent confounders using marginal structural networks. Marginal structural networks have been used in epidemiology and other statistical fields, and exploit inverse probability of treatment weighting (IPTW) to adjust for bias. They require two components - the covariate-dependent probability of treatment assignment (propensity) and the treatment-dependent response. In this work, they use RNNs (LSTMs) to approximate each of these components. They use a commonly-used synthetic dataset to develop and test their model, comparing against standard marginal structural networks and a Bayesian approach. Explanation of overall score: This is a strong submission overall - the experiments are reasonably convincing, the architecture and approach is well explained, and the problem seems important. I also think the idea of bringing approaches used elsewhere (e.g. epidemiology) for causal inference to a NIPS audience, and demonstrating how neural networks can be beneficial, is a worthy endeavour. Explanation of confidence score: I am not very familiar with one of the primary comparisons in this work (the BTRC model) and therefore can’t guarantee that the comparison was fair. Quality: * The analyses include two important experiments: varying the level of time-dependent confounding (specifically, how treatment assignment depends on the patient’s current state, which is the tumour diameter in these experiments), and varying the prediction horizon (for different levels of confounding). * In the first experiment they also compare their proposed approach with a hybrid where the IPTWs are derived from the “standard” MSM model. Further along this vein, I would be interested to know how well a “vanilla” RNN would perform at this task. That is, if the task in Figure 4 is simply to predict the treatment response at the next time-point, how well can you do if you neglect the MSM modelling entirely, and treat it as a standard regression problem? I think, for readers not familiar with MSMs, this comparison might help to motivate why the MSM framework is necessary. * Equivalently, it would be interesting to see how well a MSM performs if the IPTWs are estimated using RNNs, but the treatment response itself is given by the standard regression model. In essence, this work proposes an upgrade of MSMs with two new parts, and demonstrates that this works better. But then the question is where did the greatest performance gain come from? * I am slightly confused by the performance in Figure 4 when there is no time-dependent confounding. In this case, shouldn’t all IPTWs (if estimated correctly) be the same? If this is true, then shouldn’t the performance of the R-MSN and the Seq2Seq/Logistic baselines be the same? My intuition in this case is that the RMSE should be driven by the baseline difficulty to predict if a treatment will be assigned (since, according to (9), gamma = 0 indicates random assignments), on top of the error of the treatment-dependent response prediction. If the latter part uses the same model, then how does the RMSE differ? * To thoroughly demonstrate the power of the RNNs to adapt to different types of response models, I would have liked to see experiments on a different dataset. Clarity: * The submission is well-organised and fairly clearly written. My biggest complaint would be that there is relatively less detail about how the final model specifically looks - it would perhaps be sufficient to refer back to equation (4) after the R-MSN has been defined (around equation (5)), to make it clear how the RNNs are fitting in to the MSM model. * There are some small errors in the text of 4.3 - in Step 2, it says Figure 3(c) instead of Figure 3(b), and later says “decoder” when it should probably say “encoder”. (I only point these out because these descriptions are quite important!). * As far as I know, there is no limit on the length of the references for the main paper, so I would recommend to move all references appearing in the main text to the main bibliography, it’s a little confusing otherwise. * The appendix contains a good amount of detail about training and hyperparameter optimisation, making this paper likely easy to reproduce 🙂 (I hope the authors will also provide code when feasible) Originality: * To the best of my knowledge the idea to parametrise elements of a MSM using RNNs is original. * The appendix contains an extended related work section, helpfully placing the work in context. Significance: * The problem of accounting for time-dependent confounders is important in various domains of applied machine learning. * This idea to me seems like a nice application of methods from deep learning towards more traditional statistical sciences, which may also help to publicise models such as MSMs to the NIPS community.

Reviewer 3



After reading the rebuttal and others reviews, I have decided to upgrade the score. The added seq2seq results and the argument on the use of simulation data are convincing. The paper proposes a new recurrent neural network architecture to forecast treatment response called R-MSM. which contains 2 submodules: a propensity network that learns the probability of each treatment and a seq2seq model that predict the treatment outcome. + The problem of predicting the patients' treatment outcome is interesting. + The idea of using the Inverse Probability of Treatment Weighting (IPTW) to adjust for time-dependent confounding in the objective function is new. - The model architecture is too specific to the application. It is not clear how the two models are combined together. It seems that the model is solving two different problems separately. - The experiments are not convincing. In table 1, an important baseline would be seq-2-seq RNN without the propensity network. Furthermore, the data is only from the simulation and it is not clear whether the method could work for real-world data. - Typos: line 130, use->used, line 264 typo in description, line 205, continuos -> continuous, table 1 overflows